# Knockout of Purinergic P2Y_6_ Receptor Fails to Improve Liver Injury and Inflammation in Non-Alcoholic Steatohepatitis

**DOI:** 10.3390/ijms24043800

**Published:** 2023-02-14

**Authors:** Kazuhiro Nishiyama, Kohei Ariyoshi, Akiyuki Nishimura, Yuri Kato, Xinya Mi, Hitoshi Kurose, Sang Geon Kim, Motohiro Nishida

**Affiliations:** 1Graduate School of Pharmaceutical Sciences, Kyushu University, Fukuoka 812-8582, Japan; 2National Institute for Physiological Sciences (NIPS), National Institutes of Natural Sciences, Okazaki 444-8787, Japan; 3Exploratory Research Center on Life and Living Systems (ExCELLS), National Institutes of Natural Sciences, Okazaki 444-8787, Japan; 4College of Pharmacy, Dongguk University-Seoul, Goyang-si 10326, Gyeonggi-Do, Republic of Korea

**Keywords:** purinergic P2Y_6_ receptor, nonalcoholic steatohepatitis, inflammation

## Abstract

Nonalcoholic steatohepatitis (NASH) is a disease that progresses from nonalcoholic fatty liver (NAFL) and which is characterized by inflammation and fibrosis. The purinergic P2Y_6_ receptor (P2Y_6_R) is a pro-inflammatory G_q_/G_12_ family protein-coupled receptor and reportedly contributes to intestinal inflammation and cardiovascular fibrosis, but its role in liver pathogenesis is unknown. Human genomics data analysis revealed that the liver P2Y_6_R mRNA expression level is increased during the progression from NAFL to NASH, which positively correlates with inductions of C-C motif chemokine 2 (CCL2) and collagen type I α1 chain (Col1a1) mRNAs. Therefore, we examined the impact of P2Y_6_R functional deficiency in mice crossed with a NASH model using a choline-deficient, L-amino acid-defined, high-fat diet (CDAHFD). Feeding CDAHFD for 6 weeks markedly increased P2Y_6_R expression level in mouse liver, which was positively correlated with CCL2 mRNA induction. Unexpectedly, the CDAHFD treatment for 6 weeks increased liver weights with severe steatosis in both wild-type (WT) and P2Y_6_R knockout (KO) mice, while the disease marker levels such as serum AST and liver CCL2 mRNA in CDAHFD-treated P2Y_6_R KO mice were rather aggravated compared with those of CDAHFD-treated WT mice. Thus, P2Y_6_R may not contribute to the progression of liver injury, despite increased expression in NASH liver.

## 1. Introduction

Even if patients have no obvious drinking history, nonalcoholic fatty liver disease (NAFLD) is a liver illness that resembles alcoholic liver disease. The main contributing factor to the onset of NAFLD is obesity [1,2]. Nonalcoholic fatty liver (NAFL) and nonalcoholic steatohepatitis (NASH) are two symptoms of NAFLD. Triglyceride buildup is the cause of NAFL formation. NASH is an advanced type of NAFL, in which the liver tissue exhibits fibrosis comparable to alcoholic steatohepatitis and an influx of inflammatory cells [3]. It is generally accepted that inflammatory cell infiltration takes place first, followed by fibrosis, in terms of the evolution of the two conditions. According to epidemiology, NASH is linked to dyslipidemia, hypertension, fasting hyperglycemia, and metabolic syndrome [4]. The number of NAFL/NASH patients is increasing year by year, and the development of epoch-making preventive and therapeutic drugs is urgently needed [5].

Extracellular nucleosides and nucleotides are known as major damage-associated molecular patterns and mediate inflammation through stimulating a group of G-protein-coupled receptors (GPCRs) known as purinergic P2Y receptors (P2YRs) and purinergic P2X ion channels (P2XRs) [6,7]. Purinergic receptors are expressed in almost all mammalian tissues [8]. These receptors are also expressed in liver resident cells and play a critical role in maintaining liver function [9,10]. Based on their G-protein selectivity and sequence similarity, P2YRs have been divided into two subfamilies: P2Y_12_-like receptors (P2Y_12_R, P2Y_13_R, and P2Y_14_R) share 45–50% sequence similarity and pair mostly with pertussis toxin-sensitive G_i/o_, whereas P2Y_1_-like receptors (P2Y_1_R, P2Y_2_R, P2Y_4_R, P2Y_6_R, and P2Y_11_R) share 28–52% sequence homology [11]. Among them, we previously reported that P2Y_6_R couples with G_q_ and G_12_ family proteins and takes part in the signaling cascades that lead to pressure overload-induced cardiac remodeling, particularly to interstitial fibrosis [12]. Exposure of cardiomyocytes to mechanical stress the increased expression level of P2Y_6_R mRNA, and the pharmacological inhibition of P2Y_6_R prevents pressure overload-induced heart failure in mice [12]. By contrast, cardiomyocyte-specific overexpression of P2Y_6_R exacerbates pressure overload-induced heart failure [13]. We also reported that P2Y_6_R contributes to the progression of age-related hypertension and inflammatory bowel disease, and P2Y_6_R knockout mice show the prevention of tissue remodeling, including fibrosis and inflammation [14,15]. These results suggest a positive relationship between P2Y_6_R expression and the severity of tissue fibrosis and inflammation. In addition, Gα_12_ levels are markedly diminished in liver biopsies from NAFLD patients, and Gα_12_ overexpression induced by miR-16 dysregulation contributes to liver fibrosis by promoting autophagy in hepatic stellate cells [16]. In contrast, Gα_12_ signaling is found to regulate sirtuin (SIRT) 1-dependent mitochondrial energy expenditure through hypoxia-inducible factor-1α-dependent ubiquitin-specific peptidase 22 induction, which leads to the stabilization of SIRT1 [17]. These reports strongly suggest that P2Y_6_R-Gα_12_ signaling is a plausible drug target for NASH.

In this study, we aim to clarify the involvement of P2Y_6_R signaling in the progression of NASH and to show whether P2Y_6_R becomes a new therapeutic target for NASH. Unexpectedly, however, we demonstrate that the suppression of P2Y_6_R signaling fails to improve the progression of liver injury using mice fed with the choline-deficient, L-amino acid-defined, high-fat diet (CDAHFD).

## 2. Results

### 2.1. Increase in mRNA Expression Levels of P2Y_6_R, CCL2 and Col1a1 in NASH Patients

We searched open resources to investigate whether the mRNA expression levels of P2YRs and C-C motif chemokine 2 (CCL2), as inflammatory markers, and collagen type I alpha 1 chain (Col1a1), as a fibrosis marker, are altered in NASH patients. The mRNA expression levels of CCL2 and Col1a1 were increased in NASH patients’ livers compared with NAFL patients’ livers (Figure 1A). Among P2Y receptors, P2Y_1_R, P2Y_6_R, P2Y_11_R, P2Y_13_R, and P2Y_14_R were increased in NASH patients compared with NAFL patients (Figure 1A). The mRNA expression level of P2Y_6_R showed a positive correlation with the mRNA expression levels of CCL2 and Col1a1 (Figure 1B). These data suggested that the liver P2Y_6_R mRNA expression increases during the progression from NAFL to NASH.

### 2.2. Effects of P2Y_6_R Knockout on Body Status Changes Induced by CDAHFD

Next, to investigate whether the expression levels of P2Y_6_R and CCL2 are increased in a diet-induced NASH mouse model (CDAHFD), we measured the expression levels of P2Y_6_R and CCL2 in the liver. P2Y_6_R and CCL2 mRNA expression levels were increased in the liver of CDAHFD-fed mice compared to mice fed the standard (control) diet (Figure 2A). The mRNA expression level of P2Y_6_R showed a positive correlation with the mRNA expression level of CCL2 in mice liver (Figure 2B). The protein expression level of P2Y_6_R was increased in hepatocytes from mice fed with CDAHFD compared to those fed the control diet (Figure 2C). We examined the impact of P2Y_6_R functional deficiency in the diet-induced NASH model. It has been reported that mice fed with a normal diet gain body weight, while mice fed with CDAHFD maintain little or even slightly lose body weight [18]. CDAHFD-fed WT mice slightly lost body weight. The body weight of P2Y_6_R knockout (KO) mice fed with CDAHFD was changed in the same manner as occurred for WT mice fed with CDAHFD (Figure 2D). We confirmed that the food intake, water intake, and stool numbers in P2Y_6_R KO mice were similar to those in WT. Urine volume in P2Y_6_R KO mice fed with CDAHFD was reduced compared to WT mice fed with CDAHFD (Figure 2E–H).

### 2.3. Liver Weights and Serum Levels of AST and ALT

CDAHFD feeding increased liver weight in WT mice (Figure 3A). Liver weights in CDAHFD-fed P2Y_6_R KO mice were reduced in comparison to those in CDAHFD-fed WT mice (Figure 3A). To evaluate liver damage, we measured serum levels of liver-damaging enzymes such as aspartate aminotransferase (AST) and alanine aminotransferase (ALT) [19]. AST in P2Y_6_R KO mice fed with a CDAHFD diet was increased in comparison to WT mice fed with CDAHFD (Figure 3B). ALT in P2Y_6_R KO mice fed with a CDAHFD diet was increased in comparison to P2Y_6_R KO mice fed with the control diet (Figure 3C). These data suggested that the knockout of the P2Y_6_R aggravates CDAHFD-induced liver injury.

### 2.4. Liver Histology

We found that CDAHFD feeding caused predominantly middle droplet steatosis and inflammatory cell infiltration in the liver of WT mice and P2Y_6_R (Figure 4A). The severity of steatosis was similar in WT and P2Y_6_R KO mice fed with CDAHFD (Figure 4B). The infiltration of inflammatory cells was similar in CDAHFD-fed WT and P2Y_6_R KO mice (Figure 4A). These data suggested that a knockout of the P2Y_6_R does not affect CDAHFD-induced steatosis.

### 2.5. Inflammation and Fibrosis of the Liver

CCL2 and interleukin-6 (IL-6) were analyzed as factors involved in the inflammation of the liver [20]. The mRNA expression level of CCL2, but not IL-6, in the liver of CDAHFD-fed P2Y_6_R KO mice was increased more than in the liver of CDAHFD-fed WT mice (Figure 5A,B). TGFβ1 and Col1a1 were analyzed as factors involved in liver fibrosis. CDAHFD feeding increased the expression levels of both fibrotic factors in the liver of WT mice and P2Y_6_R mice. Both TGFβ1 and Col1a1 were similarly expressed in WT and P2Y_6_R KO mice fed with CDAHFD (Figure 5C,D). These data suggested that a knockout of the P2Y_6_R aggravates CDAHFD-induced inflammation but not fibrosis.

## 3. Discussion

P2Y_6_R is expressed in various cell types. P2Y_6_R KO mice are alive and do not differ in growth or fertility from its littermate WT mice, but show remarkable phenotypes in several stress conditions [21]. The synthesis of inositol 1,4,5-trisphosphate induced by UDP stimulation was eliminated in thioglycolate-elicited mouse macrophages, demonstrating that P2Y_6_R is the only UDP-responsive receptor present in macrophages. UDP-dependent enhanced responsiveness to LPS stimulation was abolished in P2Y_6_R KO macrophages [21]. Endothelial-dependent vasorelaxation, induced by UDP stimulation, and contraction, induced by UDP upon the inhibition of endothelial nitric oxide synthase, were abolished in P2Y_6_R-deficient aortas [21]. P2Y_6_R KO mice did not display atherosclerosis, vascular inflammation, age-related hypertension and intestinal inflammation [14,15,22]. Compared to wild-type mice, P2Y_6_R KO mice show a significant resistance to the skin papilloma-inducing effects of 7,12-dimethylbenz[a]anthracene/12-O-tetradecanoylphorbol-13-acetate [23]. P2Y_6_R deletion prevented the neuronal and memory loss caused by tubulin-associated unit causes [24].

Hepatocytes, immune cells such as Kupffer cells and macrophages, and hepatic stellate cells are involved in the progression of NASH [25]. According to the database [26,27], P2Y_6_R is highly expressed in immune cells such as macrophages. P2Y_6_R regulates phagocytosis and the production of inflammatory cytokines in macrophages or microglia [22,28,29]. P2Y_6_R is also expressed in hepatic stellate cells (HSCs) [30]. Treatment of activated HSCs with UDP (native P2Y_6_R agonist) tripled the mRNA levels of procollagen-1 [30]. The inhibition of P2Y_6_R ameliorates the pathology of the alcoholic steatohepatitis (ASH) model [31]. P2Y_6_R deficiency in adipocytes protects mice from diet-induced obesity due to enhanced energy expenditure and reduced inflammation, with white adipose tissue browning also being reported [32]. These reports suggest that P2Y_6_R may contribute to the progression of liver inflammation and fibrosis. However, we demonstrated that the degree of liver fibrosis in P2Y_6_R KO mice fed with CDAHFD was similar to that in WT mice with CDAHFD, but inflammation and liver injury were rather exacerbated in P2Y_6_R KO mice. P2Y_6_R KO mice have been shown to exhibit different phenotypes depending on the environment and experimental site, even with the same inflammation model [15,33]. In addition, knockout of P2Y_6_R in muscle has been shown to increase insulin resistance [32]. Liver weights in P2Y_6_R KO mice fed with CDAHFD were reduced in comparison to WT mice fed with CDAHFD. The liver weight increases in proportion to the severity of fatty liver present in the CDAHFD model, but it has been reported that the liver weight decreases when fibrosis also progresses [3,34]. The mRNA expression level of CCL2 in P2Y_6_R KO mice fed with CDAHFD was increased compared to that in WT mice fed with CDAHFD. These data suggest that P2Y_6_R knockout slightly promotes liver fibrosis and inflammation. Urine volume in P2Y_6_R KO mice fed with CDAHFD was reduced compared to that in WT mice fed with CDAHFD. It has been reported that P2Y_6_R KO mice display more frequent micturition, with smaller bladder capacity compared to WT mice [35]. P2Y_6_R is also expressed in the kidney [26,27], but its physiological role is unclear. In addition, P2Y_6_R contributes to the pathogenesis of heart failure [12,13] and hypertension [14]. CDAHFD may stimulate P2Y_6_R signaling in the cardiovascular and renal tissues, causing the difference in urine output between P2Y_6_R KO and WT. Future investigations using tissue-specific P2Y_6_R KO mice are necessary to identify which cells are regulated by P2Y_6_R in the pathogenesis of NASH.

We showed that among the P2Y subtypes, P2Y_1_R, P2Y_6_R, P2Y_11_R, P2Y_13_R, and P2Y_14_R are increased in patients with NASH compared with patients with NAFL. Among them, P2Y_6_R showed the best positive correlation with CCL2 and Col1a1. These data suggest that P2Y_6_R expression level is increased according to the degree of inflammation and fibrosis. Infiltration of inflammatory cells with high P2Y_6_R expression may contribute to elevated P2Y_6_R expression in the NASH liver. Since the expression of P2Y_6_R is also increased by LPS [36], its expression is regulated in response to inflammation at the transcription level. On the other hand, we previously reported that P2Y_6_R proteins are internalized and degraded by cysteine modification by electrophiles and nitric oxide [15]. Oxidative stress is also known to contribute to the pathogenesis of NASH [4]. In the future, it is necessary to confirm the expression level and localization of P2Y_6_R at the protein level, including not only regulation of transcriptional level but also post-translational modification in patients, to clarify the role of P2Y_6_R in NASH pathogenesis.

Smoking is positively correlated with NAFLD [37]. Involvement of intestinal nicotine concentration and intestinal microbiota has been reported [37]. On the other hand, it is also known that smoking components contain large amounts of electrophilic substances, such as aldehydes [38]. It is also known that smoking causes oxidative stress. Therefore, the internalization and degradation of P2Y_6_R by smoking may exacerbate the pathology of NASH. Conversely, there is an epidemiological report that cigarette smoke suppresses the risk of onset, progression, and recurrence of ulcerative colitis (UC) [39]. We recently reported that post-translational modification of cysteine in the intracellular 3rd loop of P2Y_6_R through covalent binding with electrophilic molecules promotes redox-dependent (β-arrestin-independent) P2Y_6_R internalization, which contributes to the attenuation of UC progression in mice [15]. As smoking aldehydes are environmental electrophiles, smoking aldehydes may increase the risk of NAFLD severity by promoting internalization and degradation of P2Y_6_R.

P2YRs are GPCRs that have been further divided into two subfamilies based on G-protein selectivity and sequence similarity. Among the P2YRs, P2Y_2_R is known to promote high-fat diet (HFD)-induced hepatic steatosis [40]. Platelets control liver tumor growth through P2Y_12_R-dependent CD40L release in NAFLD [41]. Another study revealed that mice lacking P2Y_14_R selectively in adipocytes were protected from obesity and displayed reduced liver weight compared to HFD control mice [42]. Reduced obesity and hepatic steatosis further contributed to improved insulin sensitivity in the liver of adipocyte-P2Y_14_R KO mice [42]. On the other hand, P2Y_6_R is known to couple with G_12/13_. G_12_-mediated signaling is known to regulate hepatic lipid metabolism, and G_12_-knockout mice are reported to display aggravated fatty liver [17]. We showed that P2Y_6_R signaling, despite its elevated expression, is not involved in pathogenesis and production of inflammatory markers in fatty liver. P2Y_6_R may maintain liver homeostasis through G_12_ signaling.

Other purinergic receptors, namely P2XRs and adenosine receptors (ARs), have also been reportedly involved in NASH/NALFD progression. P2X7R expression level was elevated in hepatocytes, Kupffer cells, and liver sinusoidal endothelial cells of NASH model mice [43]. In mice given carbon tetrachloride (CCl4) and HFD, the deficiency of P2X_7_R prevents inflammation, fibrosis and hepatocyte apoptosis [43,44]. P2X_7_R activation on Kupffer cells enhances TNFα and monocyte chemotactic protein-2 (MCP-2) production in HFD mice treated with CCl4 [44]. These findings imply that P2X_7_R antagonism might be a beneficial strategy for the treatment of NASH. Adenosine A_1_AR expression in the liver of streptozotocin (STZ)-induced diabetic rats was elevated [45]. A separate study, however, asserted that A_1_AR expression remained unchanged while A_2A_AR and A_3A_R receptor levels were dramatically elevated in STZ-treated rat liver [46]. A_2A_AR activation reduces inflammation [47,48], but its absence increases pro-inflammatory responses [49]. Furthermore, in mice, the absence of whole-body A_2A_AR exacerbated HFD-induced NAFLD and hepatic inflammation [50]. As a result, A_2A_AR deficiency in hepatocytes and macrophages contributed to increased inflammation [50]. The anti-inflammatory activity of A_2A_AR was also observed in a study of the methionine- and choline-deficient (MCD)-induced NASH animal model. The A_2A_AR KO mice, treated with the MCD-NASH model show a larger body weight, increased liver inflammation, and more severe hepatic steatosis than the control mice [51]. The biological effect of A_2A_R activation in reducing inflammation caused by lipotoxicity can lead to the protection of mouse liver against NASH progression [52,53]. A_2B_AR has also been shown to have an important function in the regulation of fatty liver disease. A_2B_AR deficiency protected mice from hepatic steatosis and the formation of fatty liver [54]. In diabetic KKA^Y^ mice, inhibiting A_2B_AR with the specific antagonist ATL-801 reduced glucose production during hyperinsulinemic-euglycemic clamp tests [55]. According to some studies, the activation of A_2B_AR suppressed lipogenic genes such as sterol regulatory element-binding protein-1 (SREBP-1). The A_2B_AR KO mice fed with HFD caused hepatic steatosis with increased plasma triglyceride and cholesterol levels [56]. Furthermore, hepatic A_2B_AR overexpression and activation lowered lipid production in the liver and enhanced whole-body metabolism [56]. On a typical diet, A_2B_AR KO mice lost weight and had enhanced de novo lipogenesis, resulting in raised liver triglyceride levels. Increased glucokinase and fatty acid synthase mRNA levels revealed poor lipid metabolism in the liver of A_2B_AR KO mice [57]. The A_2B_AR KO mice fed with HFD show poor glucose tolerance and insulin sensitivities [58]. WT mice treated with an A_2B_AR agonist/partial agonist, BAY60-6553, reportedly enhance glucose and insulin tolerance as well as lower fasting blood glucose levels [58]. These reports suggest that adenosine signaling through A_2A_AR and/or A_2B_AR activations will be a promising therapeutic target for the treatment of liver disorders.

Recent research has underlined the significance of the A_3_AR in NAFLD/NASH. A_3_AR expression was reduced 1.9-fold in NAFLD patients’ livers compared to controls, indicating possible involvement by the receptor in NAFLD pathogenesis [59]. A_3_AR deficiency in mice given an HFD increased the expression of genes implicated in hepatic inflammation and steatosis [59]. The researchers demonstrated that an A3_A_R agonist prodrug (MRS7476) protected the STAM mouse model from the development of NASH [59]. MRS7476’s two succinyl ester groups significantly improve its water solubility and are most likely cleaved in the gut rather than at the site of action. Another study found that the A_3_AR agonist Cl-IB-MECA (namodenoson) was effective in treating NASH in mice [60]. The drug namodenoson is currently in Phase 2 clinical trials for NASH therapeutics [10].

In this study, we show that P2Y_6_R knockout slightly promotes NASH pathology. However, we have not investigated whether P2Y_6_R activation inhibits the progression of NASH. UDPβS and Up3U are likewise P2Y_6_R agonists [61]. Several potent and selective agonists for this receptor have been discovered, including 3-phenacyl-UDP (PSB-0474), 5-iodo-UDP (MRS2693), α,β-methylene-UDP (MRS2782), INS48823, and 5-O-methyl-UDPA [61]. Several analogues of boranophosphates are significantly more active at the P2Y_6_R [61]. In the future, it will be necessary to examine whether such various P2Y_6_R agonists suppress the pathological progression of NASH.

A limitation of this study is that almost all experiments were analyzed using NASH/NAFLD model mice, when the role of P2Y_6_R in human NASH pathogenesis might in fact differ from that in mice. In this study, the CDAHFD model was used as the NASH model [18]. One of the most frequently utilized techniques in NASH research is the MCD model, which involves providing a diet low in both methionine and choline. Because methionine and choline are required for the hepatic secretion of triglycerides in the form of very low-density lipoproteins (VLDL), lipid export from the liver to peripheral tissues may be impaired in these models due to the defective incorporation of triglycerides into apolipoprotein B (ApoB), or reduced ApoB synthesis or excretion. The phenotype caused by the MCD diet includes macrovesicular steatosis, hepatocellular apoptosis, inflammation, oxidative stress, and fibrosis [62]. One disadvantage for using the MCD model to promote NASH development is that these mice will cause a significant systemic weight loss [63,64]. CDAHFD model may be a mouse model of rapidly progressive liver fibrosis and could be potentially useful for better understanding human NASH disease as well as in the development of efficient therapies for this condition. However, the pathogenesis and etiology of CDAHFD-fed mice do not necessarily match those of human patients with NASH. Future studies using induced pluripotent stem cell (iPS cell) and organoids will be necessary for elucidating the role(s) of purinergic signaling in human NASH.

In conclusion, we demonstrated using an online database that P2Y_6_R expression level is increased in NAFLD patients according to the degree of inflammation and fibrosis. We also revealed using NASH model mice that knockout of the P2Y_6_R fails to improve (but instead aggravates) liver injury and inflammation. Our findings will prompt reconsideration of P2Y_6_R-G_12_ signaling as a therapeutic strategy for NASH.

## 4. Materials and Methods

### 4.1. GEO Datasets Analysis

GEO RNA-Sequencing Experiments Interactive Navigator [65] was used to retrieve normalized transcript levels from the GEO dataset GSE167523 [66]. We analyzed the transcript levels of P2YRs, CCL2 and Col1a1 in NAFL and NASH patients.

### 4.2. Animals

All animal care and experimental procedures used in this study were approved by the ethics committees at the Animal Care and Use Committee of Kyushu University. Systemic P2Y_6_R KO mice were backcrossed onto C57BL/6J mice background, as described previously [13,14,15]. We used P2Y_6_R KO and cohoused littermates as WT (8–11 weeks old, male). Animals were maintained under a 12 h/12 h light/dark cycle.

### 4.3. Diet-Induced NASH Model

We fed male mice CDAHFD (Research Diets Inc., New Brunswick, NJ, USA, Cat# A06071302) or a control diet (Research Diets Inc., New Brunswick, NJ, USA, Cat# A06071314) for 6 weeks [18]. After 6 weeks, we euthanized all mice under isoflurane anesthesia. We took the liver from mice and measured liver weights. We collected blood samples from the caudal vena cava and centrifuged these at 10,000× *g* for 10 min.

### 4.4. Immunohistochemistry

For tissue sections, the liver tissue was fixed by 4% paraformaldehyde. Frozen sections (5 μm thick) were cut and prepared for immunofluorescent staining [67]. The expression of P2Y_6_R was detected using rabbit anti-P2Y_6_R antibodies (#APR011: Alomone Labs, Jerusalem, Israel). The immunoreactivity of P2Y_6_R was detected using an Alexa Fluor 488-labeled goat anti-rabbit IgG antibody (#A-11008: ThermoFisher Scientific, Waltham, MA, USA). Non-specific immunoreactivity was blocked with 10% normal goat serum, 1% BSA, and 0.3% Triton X-100 in PBS. After incubation with the secondary antibody, images were captured using a confocal laser-scanning microscope (LSM900, Zeiss, Oberkochen, Germany).

### 4.5. Serum Biochemical Analysis

We measured serum levels of AST and ALT by using Fuji Dry-Chem NX5000 (FUJIFILM Medical, Tokyo, Japan) [2].

### 4.6. Histological Evaluation of the Liver

We fixed the liver with 10% neutral buffered formalin and embedded these in paraffin. We performed hematoxylin and eosin (H&E) staining, as previously described [2]. We quantified empty envelopes using ImageJ and evaluated as steatosis [2].

### 4.7. RNA Isolation and qPCR

We extracted total RNA from the liver, as previously described [67]. The RNA was used to synthesize complementary DNA using ReverTra Ace (Toyobo, Osaka, Japan). We performed qPCR as previously described [67,68]. Primer sequences used are summarized in Appendix A. To normalize cDNA levels, 18 s rRNA expression was used as an endogenous control.

### 4.8. Statistics

Statistical analysis were performed by using GraphPad Prism 9.0 (GraphPad Software, LaJolla, CA, USA) [2]. All results were expressed as mean ± SEM from at least 3 independent experiments. Statistical comparisons were determined using two-tailed Student’s *t* tests (for two groups) or using the one-way ANOVA method with Tukey’s post hoc test (for three or more groups).

## Figures and Tables

**Figure 1 ijms-24-03800-f001:**
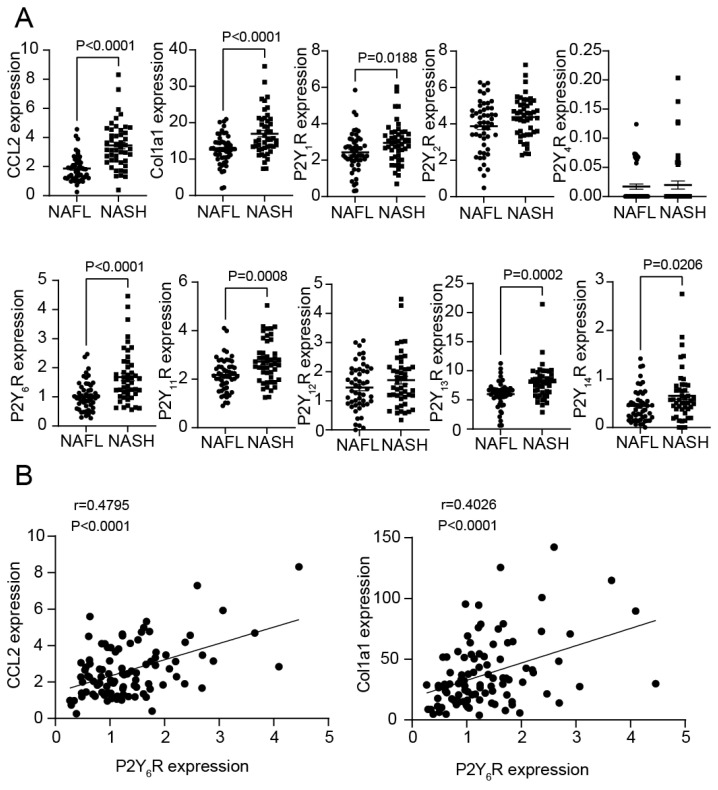
Increase in mRNA expression levels of P2Y_6_R, CCL2 and Col1a1 in NASH patients. (**A**) The expression of CCL2, Col1a1, and P2Y receptors was analyzed in a Gene Expression Omnibus (GEO) dataset (GSE167523) containing the expression profile of the liver from NASH patients. (**B**) Correlation diagram between the expression levels of P2Y_6_R and the expression levels of CCL2 and col1a1. Data are shown as the mean ± SEM. (NAFL; *n* = 51, NASH; *n* = 47) Student’s *t* test (**A**). Pearson’s product moment correlation coefficient (**B**).

**Figure 2 ijms-24-03800-f002:**
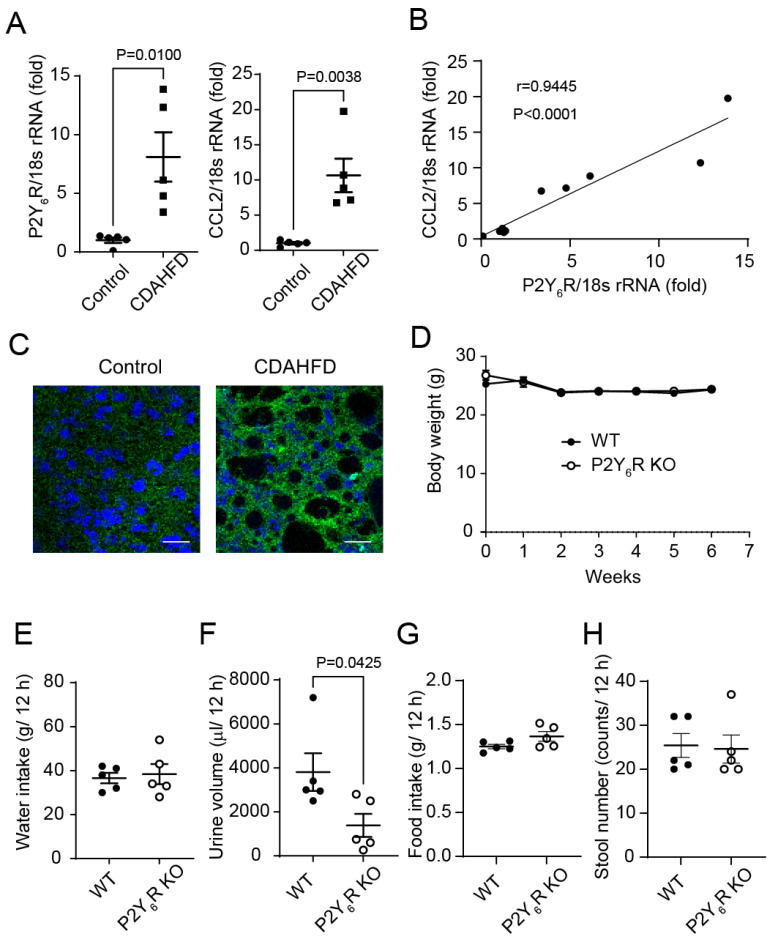
Effects of P2Y_6_R knockout on body status changes induced by CDAHFD. (**A**) The mRNA expression of P2Y_6_R and CCL2 in the livers of C57BL/6J mice fed with standard diet (control) or CDAHFD for 6 weeks (*n* = 5 in each group). (**B**) Correlation diagram between the expression levels of P2Y_6_R and the expression levels of CCL2 in mouse liver. (**C**) Immunohistological staining of P2Y_6_R in the liver of mice fed with standard diet or CDAHFD for 6 weeks. Scale bar: 40 μm. CDAHFD were fed to WT and P2Y_6_R KO mice for 6 weeks. (**D**) Body weight. (**E**) Water intake. (**F**) Urine volume. (**G**) Food intake. (**H**) Stool number. Data are shown as the mean ± SEM (*n* = 5 in each group). Student’s *t* test (**A**,**F**). Pearson’s product moment correlation coefficient (**B**).

**Figure 3 ijms-24-03800-f003:**
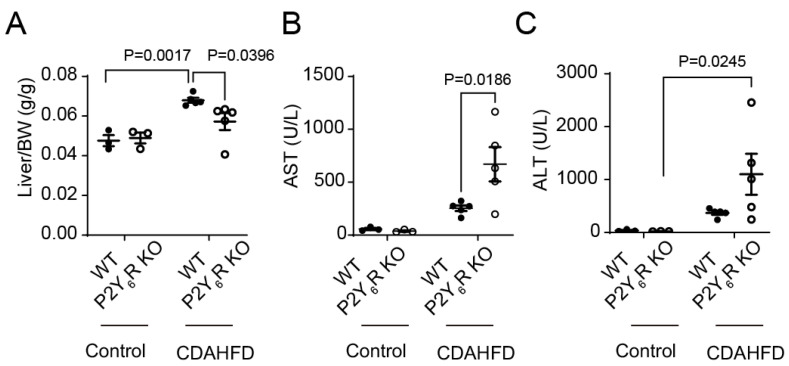
Liver weights and serum levels of AST and ALT in WT and P2Y_6_R KO mice fed with CDAHFD. (**A**) Liver weights. (**B**,**C**) Comparison of serum levels of AST (**B**) and ALT (**C**) among P2Y_6_R KO and WT mice. Data are shown as the mean ± SEM (control WT: *n* = 3, control P2Y_6_R KO: *n* = 3, CDAHFD WT: *n* = 5, CDAHFD P2Y_6_R KO: *n* = 5). Two-way ANOVA followed Sidak’s multiple comparison test.

**Figure 4 ijms-24-03800-f004:**
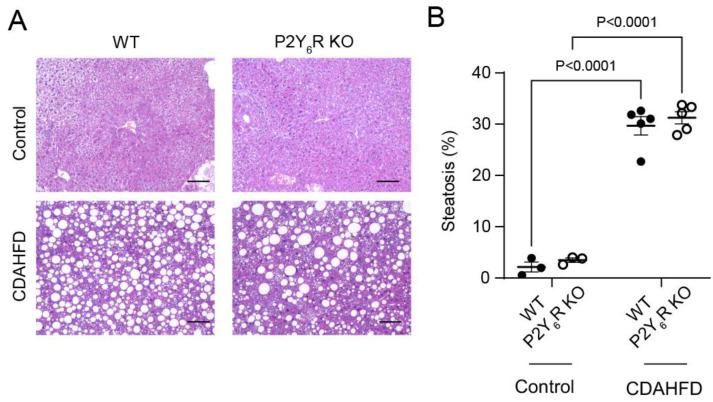
Effects of P2Y_6_R knockout on liver steatosis induced by CDAHFD. (**A**) H&E-stained images of liver sections. Scale bar: 100 μm. (**B**) Liver steatosis. Data are shown as the mean ± SEM (control WT: *n* = 3, control P2Y_6_R KO: *n* = 3, CDAHFD WT: *n* = 5, CDAHFD P2Y_6_R KO: *n* = 5). Two-way ANOVA followed Sidak’s multiple comparison test.

**Figure 5 ijms-24-03800-f005:**
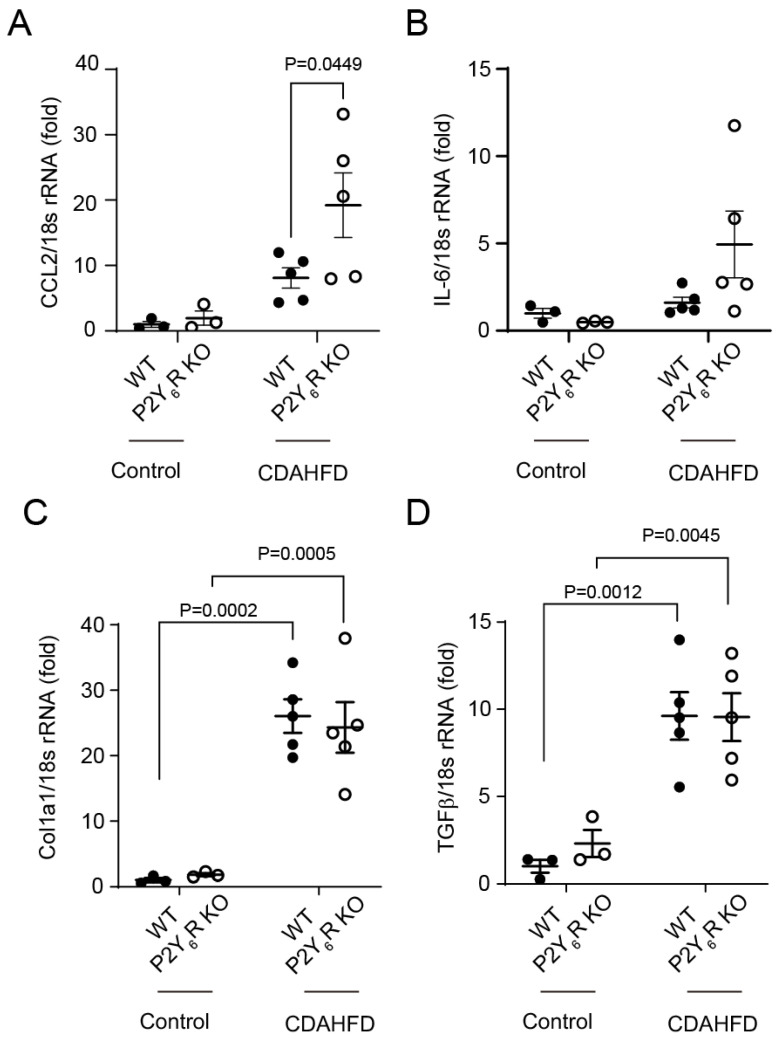
Knockout of P2Y_6_R accelerated expression of liver inflammation markers induced by CDAHFD feeding. Expression of CCL2 (**A**), IL-6 (**B**), Col1a1 (**C**), and TGFβ (**D**) in liver were quantified by real-time qPCR. Data are shown as the mean ± SEM (control WT: *n* = 3, control P2Y_6_R KO: *n* = 3, CDAHFD WT: *n* = 5, CDAHFD P2Y_6_R KO: *n* = 5). Two-way ANOVA followed Sidak’s multiple comparison test.

## Data Availability

Not applicable.

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
