# Peer review of "Knockout of Purinergic P2Y6 Receptor Fails to Improve Liver Injury and Inflammation in Non-Alcoholic Steatohepatitis"

_ijms, 2023, doi:10.3390/ijms24043800_

Round 1

Reviewer 1 Report

Journal – IJMS (ISSN 1422-0067)

Manuscript  ijms- 2154491

Type research article 

Title.  Knockout of purinergic P2Y6 receptor fails to improve liver in jury and inflammation in non-alcoholic steatohepatitis

Authors Kazuhiro Nishiyama, Kohei Ariyoshi, Akiyuki Nishimura, Yuri Kato, Xinya Mi, Hitoshi Kurose, Sang Geon Kim and Motohiro Nishida

This is an interesting work focused on evaluating functionally in a mouse model the role of the purinergic receptor P2Y6R in the progression of nonalcoholic fatty liver disease to Nonalcoholic steatohepatitis. P2Y6R knockout (KO) mice were employed as animal model. Contrary to what has been hypothesized, P2Y6R may not contribute to the progression liver injury. Although the reported negative findings, the study undoubtedly increase our knowledge on the function of Py receptors, and in particular P2Y6 receptor in the liver physiology in general and in the the aetiology of liver diseases. Please see below my comments:  

Main comment 1 – Study limitations should be included in the discussion, before the conclusions.

Minor

1. Please separate the citations from the words throughout the text

2. The sentence in lines 46-47 is lacking in supporting references, please include this work on the epidemiology of NAFL/NASH disease (DOI: 10.1016/j.jhep.2019.06.021)

3. The sentence on purinergic receptors in lines 48-51 is lacking in supporting references. Authors are kindly encouraged to cite the following reviews on PY (PMID: 32037507) and PX (DOI: 10.3390/cancers14051116 ) receptors. 

4. Lines 51-52 Purinergic receptors are expressed in almost all mammalian tissues https://www.nature.com/articles/s41598-021-91989-4 

5. Please include citations in the 4.4, 4.5, 4.6 and 4.8 sections

6. Please include the number of patients whose gene expression data were extrapolated from  online databases and analysed. IN addition more information on the computational analysis should be included in the methods 

7. Line 101 “mouse model”

8. A reference for the statement reported in line 128 should be included. Same comment, lines 153-154.

9. Line 170 “cell types”

10. For a better reading, I kindly suggest to not mention figures in the discussion

11. The fact that data were retrieved from an online database should be underlined in lines 217-218

Author Response

Responses to the Editor and Reviewers

First of all, we truly appreciate your great efforts to review our manuscript thoughtfully. We have addressed the concerns raised by the reviewers by adding explanations to each of the queries. Our response to each point is presented below and we denote where modifications to the text in the manuscript have been made.

Responses to Reviewer 1.

Main comment 1 – Study limitations should be included in the discussion, before the conclusions.

[Response]

Thank you for your comment. A limitation of this study is that almost all experiments are analyzes using NASH/NAFLD model mice., and the role of P2Y6R in human NASH pathogenesis might differ from that in mice. In this study, the CDAHFD model was used as the NASH model. One of the most often utilized techniques in NASH research is the MCD model, which involves providing a diet low in both methionine and choline. Because methionine and choline are required for hepatic secretion of triglycerides in the form of very low-density lipoproteins (VLDL), lipid export from the liver to peripheral tissues may be impaired in these models due to defective incorporation of triglycerides into apolipopro-tein B (ApoB) or reduced ApoB synthesis or excretion. The phenotype caused by the MCD diet includes macrovesicular steatosis, hepatocellular apoptosis, inflammation, oxidative stress, and fibrosis. One disadvantage for using MCD to promote NASH development is that these mice will cause a significant systemic weight loss. CDAHFD model may be a mouse model of rapidly progressive liver fibrosis and be potentially useful for better understanding human NASH disease and in the development of efficient therapies for this condition. However, the pathogenesis and etiology of CDAHFD-fed mice do not necessarily match those of human NASH. Future study using induced pluripotent stem cell (iPS cell) and organoids will be necessary for elucidating the role(s) of purinergic sig-naling in human NASH. We described study limitations in the Discussion (yellow highlight).

Minor

  1. Please separate the citations from the words throughout the text

[Response]

Thank you for your comment. We prepared the manuscript according to the IJMS format. We separated the citations from the words.

  1. The sentence in lines 46-47 is lacking in supporting references, please include this work on the epidemiology of NAFL/NASH disease (DOI: 10.1016/j.jhep.2019.06.021)

[Response]

Thank you for your comment. We cited the paper (Younossi et al. J Hepatol 2019) in the Introduction (yellow highlight).

  1. The sentence on purinergic receptors in lines 48-51 is lacking in supporting references. Authors are kindly encouraged to cite the following reviews on PY (PMID: 32037507) and PX (DOI: 10.3390/cancers14051116 ) receptors. 

[Response]

Thank you for your comment. We cited the paper (Jacobson et al. Br J Pharmacol 2020; Rotondo et al. Cancers 2022) in the Introduction (yellow highlight).

  1. Lines 51-52 Purinergic receptors are expressed in almost all mammalian tissues https://www.nature.com/articles/s41598-021-91989-4

[Response]

Thank you for your comment. We described and cited the paper (Li et al. Sci Rep 2021) in the Introduction (yellow highlight).

  1. Please include citations in the 4.4, 4.5, 4.6 and 4.8 sections

[Response]

Thank you for your comment. We cited the papers (Nishiyama et al. Journal of Cellular Physiology 2019; Nishiyama et al. Biological and Pharmaceutical Bulletin 2021) in the 4.4, 4.5, 4.6 and 4.8 sections (yellow highlight).

  1. Please include the number of patients whose gene expression data were extrapolated from online databases and analysed. IN addition more information on the computational analysis should be included in the methods 

[Response]

Thank you for your comment. We analyzed transcript levels of P2YRs, CCL2 and Col1a1 in NAFL and NASH patients (NAFL; n=51, NASH; n=47). We described more information in the legends and methods (yellow highlight).

  1. Line 101 “mouse model”

[Response]

Thank you for your comment. We changed the phrase from “model” to “mouse model” in the Results (yellow highlight).

  1. A reference for the statement reported in line 128 should be included. Same comment, lines 153-154.

[Response]

Thank you for your comment. We cited the papers (Ozer et al. Toxicology 2008; Haukeland et al. Journal of Hepatology 2006) in the Results (yellow highlight).

  1. Line 170 “cell types”

[Response]

Thank you for your comment. We changed the phrase from “cell” to “cell types” in the Discussion (yellow highlight).

  1. For a better reading, I kindly suggest to not mention figures in the discussion

[Response]

Thank you for your comment. We removed the mention of figures in the Discussion.

  1. The fact that data were retrieved from an online database should be underlined in lines 217-218

[Response]

Thank you for your comment. We demonstrated that P2Y6R expression level is increased in NAFLD patients according to the degree of inflammation and fibrosis using an online database. We changed and underlined the words in the Discussion (yellow highlight).

Reviewer 2 Report

The manuscript reports on research aimed at elucidating the role of P2Y6R in pathogenesis of nonalcoholic steatohepatitis (NASH). The authors analyzed open resource GEO datasets and found that mRNA levels of P2Y6R and chemokine CCL2 were upregulated correlatively in the liver during the course of progression from nonalcoholic fatty liver (NAFL) to nonalcoholic steatohepatitis (NASH). Based on such evidence, the authors generated NASH models in wild type and P2Y6R receptor-deficient mice by feeding a choline-deficient, L-amino acid defined, high-fat diet (CDAHFD), and compared the NASH pathogenesis. Results showed that P2Y6R and CCL2 mRNA expression were upregulated in NASH livers of wild-type mice as similar to results from human disease data analyses. However, P2Y6R deficiency unexpectedly exacerbated, rather than inhibited, CCL2 gene expression and hepatitis markers such as AST and ATL. The experiments were well designed to enable the readers to learn the unexpected role of P2Y6R in NASH pathogenesis. The article seems to me complete and worthy of publication, however, there are some concerns to be clarified.

1.      Does the elevated P2Y6R expression (Figure 2A) in fatty liver occur in liver cells? The results of immunohistochemistry in Figure 2C were explained that the increase in P2Y6R expression occurred in hepatocytes (lines 103-105), but the image was not clear enough to understand. It would be better to include the counterstained image as well like Figure 4A.

2.  In relation to above, the authors addressed that inflammatory cell infiltration was observed in the fatty liver of wild type mice (lines 141-143). Was there any change in inflammatory cell infiltration in the NASH liver in P2Y6R-deficient mice? Also, are these inflammatory cells involved in the upregulation of P2Y6R in NASH liver?

3. The results of this study clearly demonstrate that P2Y6R signaling, despite its elevated expression, is not involved in pathogenesis and production of inflammatory markers in fatty liver. This point should be stated more clearly. In addition, it is helpful to include the authors interpret why loss of P2Y6R signaling enhances fatty liver injury.

4. NASH model mice may lose weight, but not significantly. How about adding the results of giving standard food as well? This article must be usable not only to the experts of NASH, but also to the authors of other fields of medicine. For this reason, it would be better to provide general informatin on CDAHFD diet-induced NASH model, that lead to fatty liver without increasing body weight.

5. The authors should discuss about liver weight increase was inhibited in P2Y6R-deficient mice.

Author Response

Responses to the Editor and Reviewers

First of all, we truly appreciate your great efforts to review our manuscript thoughtfully. We have addressed the concerns raised by the reviewers by adding explanations to each of the queries. Our response to each point is presented below and we denote where modifications to the text in the manuscript have been made.

Responses to Reviewer 2.

  1. Does the elevated P2Y6R expression (Figure 2A) in fatty liver occur in liver cells? The results of immunohistochemistry in Figure 2C were explained that the increase in P2Y6R expression occurred in hepatocytes (lines 103-105), but the image was not clear enough to understand. It would be better to include the counterstained image as well like Figure 4A.

[Response]

Thank you for your comment. We changed to the counterstained image (Figure 2C). This image shows increased expression of P2Y6R in hepatocytes of CDAHFD-fed mice.

  1. In relation to above, the authors addressed that inflammatory cell infiltration was observed in the fatty liver of wild type mice (lines 141-143). Was there any change in inflammatory cell infiltration in the NASH liver in P2Y6R-deficient mice? Also, are these inflammatory cells involved in the upregulation of P2Y6R in NASH liver?

[Response]

Thank you for your comment. Infiltration of inflammatory cells was similar in WT and P2Y6R KO mice fed with CDAHFD. Infiltration of inflammatory cells with high P2Y6R expression may contribute to elevated P2Y6R expression in the NASH liver. This is described in the Results and Discussion (green highlight).

  1. The results of this study clearly demonstrate that P2Y6R signaling, despite its elevated expression, is not involved in pathogenesis and production of inflammatory markers in fatty liver. This point should be stated more clearly. In addition, it is helpful to include the authors interpret why loss of P2Y6R signaling enhances fatty liver injury.

[Response]

Thank you for your comment. P2Y6R is known to couple with G12/13. G12-mediated signaling is known to regulate hepatic lipid metabolism, and G12-knockout mice are reported to have aggravated fatty liver. We showed that P2Y6R signaling, despite its elevated expression, is not involved in pathogenesis and production of inflammatory markers in fatty liver. P2Y6R may maintain liver homeostasis through G12 signaling. We described that in the Discussion (green highlight).

  1. NASH model mice may lose weight, but not significantly. How about adding the results of giving standard food as well? This article must be usable not only to the experts of NASH, but also to the authors of other fields of medicine. For this reason, it would be better to provide general informatin on CDAHFD diet-induced NASH model, that lead to fatty liver without increasing body weight.

[Response]

Thank you for your comment. Mice fed a normal diet are known to gain body weight, while mice fed CDAHFD maintain little or even slightly lose body weight. WT mice fed with CDAHFD diet for 6 weeks slightly lost body weight. This is described in the Results (green highlight).

  1. The authors should discuss about liver weight increase was inhibited in P2Y6R-deficient mice.

[Response]

Thank you for your comment. Liver weights in P2Y6R KO mice fed with CDAHFD were reduced in comparison to WT mice fed with CDAHFD. In the CDAHFD model, the weight of the liver increases with fatty liver, but it has also been reported that the weight of the liver decreases when fibrosis progresses. The mRNA expression level of CCL2 in P2Y6R KO mice fed with CDAHFD was increased than in WT mice fed with CDAHFD. These data suggest that that P2Y6R knockout slightly promotes NASH pathology. This is described in the Discussion (green highlight).
